# Highly Homogeneous Current Transport in Ultra-Thin Aluminum Nitride (AlN) Epitaxial Films on Gallium Nitride (GaN) Deposited by Plasma Enhanced Atomic Layer Deposition

**DOI:** 10.3390/nano11123316

**Published:** 2021-12-07

**Authors:** Emanuela Schilirò, Filippo Giannazzo, Salvatore Di Franco, Giuseppe Greco, Patrick Fiorenza, Fabrizio Roccaforte, Paweł Prystawko, Piotr Kruszewski, Mike Leszczynski, Ildiko Cora, Béla Pécz, Zsolt Fogarassy, Raffaella Lo Nigro

**Affiliations:** 1CNR-IMM, Strada VIII, 5, 95121 Catania, Italy; emanuela.schiliro@imm.cnr.it (E.S.); salvatore.difranco@imm.cnr.it (S.D.F.); giuseppe.greco@imm.cnr.it (G.G.); patrick.fiorenza@imm.cnr.it (P.F.); fabrizio.roccaforte@imm.cnr.it (F.R.); 2Top-GaN Ltd., Sokolowska 29/37, 01-142 Warsaw, Poland; pprysta@unipress.waw.pl (P.P.); kruszew@unipress.waw.pl (P.K.); mike@unipress.waw.pl (M.L.); 3Institute of High Pressure Physics, Polish Academy of Sciences, Sokolowska 29/37, 01-142 Warsaw, Poland; 4Centre for Energy Research, Institute for Technical Physics and Materials Science Research, Konkoly-Thege, 29-33, 1121 Budapest, Hungary; cora.ildiko@energia.mta.hu (I.C.); pecz.bela@energia.mta.hu (B.P.); fogarassy.zsolt@energia.mta.hu (Z.F.)

**Keywords:** AlN, GaN, atomic layer deposition

## Abstract

This paper reports an investigation of the structural, chemical and electrical properties of ultra-thin (5 nm) aluminum nitride (AlN) films grown by plasma enhanced atomic layer deposition (PE-ALD) on gallium nitride (GaN). A uniform and conformal coverage of the GaN substrate was demonstrated by morphological analyses of as-deposited AlN films. Transmission electron microscopy (TEM) and energy dispersive spectroscopy (EDS) analyses showed a sharp epitaxial interface with GaN for the first AlN atomic layers, while a deviation from the perfect wurtzite stacking and oxygen contamination were detected in the upper part of the film. This epitaxial interface resulted in the formation of a two-dimensional electron gas (2DEG) with a sheet charge density n_s_ ≈ 1.45 × 10^12^ cm^−2^, revealed by Hg-probe capacitance–voltage (C–V) analyses. Nanoscale resolution current mapping and current–voltage (I–V) measurements by conductive atomic force microscopy (C-AFM) showed a highly homogeneous current transport through the 5 nm AlN barrier, while a uniform flat-band voltage (V_FB_ ≈ 0.3 V) for the AlN/GaN heterostructure was demonstrated by scanning capacitance microscopy (SCM). Electron transport through the AlN film was shown to follow the Fowler–Nordheim (FN) tunneling mechanism with an average barrier height of <Φ_B_> = 2.08 eV, in good agreement with the expected AlN/GaN conduction band offset.

## 1. Introduction

Due to its large and direct bandgap (6.2 eV), good thermal stability and piezoelectric properties, aluminum nitride (AlN) has been the object of significant attention for optoelectronic applications, such as ultraviolet light emitting diodes, photodetectors and sensor systems [1,2,3]. In particular, owing to the epitaxial interface with GaN and the relatively high dielectric permittivity (*κ* ≈ 8), AlN ultra-thin films have been considered as gate dielectrics in AlGaN/GaN metal insulator semiconductor–high electron mobility transistors (MIS-HEMTs) [4] and/or as passivation layers for AlGaN/GaN heterostructures, as an alternative to the conventional silicon nitride (SiN_x_) [5,6,7]. The inherent lattice mismatch, of about 2.4%, between the AlN and GaN crystal structures is responsible for a tensile strain in AlN films grown on GaN. The piezoelectric polarization associated with such tensile strain, combined with the spontaneous polarization of the AlN and GaN materials, results in the formation of a two-dimensional electron gas (2DEG) [8] at their interface, which can be exploited for the fabrication of AlN/GaN HEMTs suitable for RF applications [9,10,11,12]. Furthermore, high crystalline quality ultra-thin AlN layers on GaN have been recently employed as tunneling barriers of vertical hot electron transistors (HETs) with a graphene base, currently regarded as promising candidates for future ultra-high-frequency (THz) applications [13,14,15].

AlN thin films on GaN are typically deposited by molecular beam epitaxy (MBE) or metal organic chemical vapor deposition (MOCVD) at relatively high temperatures (>700 °C), required to obtain a high quality epitaxial interface [10,16], and a 2DEG sheet density in the order of 10^13^ cm^−2^. However, tensile-strained AlN layers with a thickness above a critical value of ~7 nm are typically subjected to relaxation or cracking phenomena during the cooling process from deposition to room temperature, due to the large thermal expansion coefficient mismatch with GaN [17,18]. More generally, such high growth temperatures can represent a serious concern in terms of process integration of AlN gate dielectrics or passivation layers in the fabrication flow of AlGaN/GaN HEMTs. In this context, the Atomic Layer Deposition (ALD) technique has been recently considered as an alternative method for the growth of thin AlN films on GaN [5,19,20] due to its unique ability to provide uniform and conformal coverage with nanometric control of the thickness, and the low process temperature in the range of 100–400 °C. Typically, trimethylaluminum (TMA) and ammonia (NH_3_) are employed as the aluminum precursor and co-reactant, respectively, and the plasma-enhanced ALD (PE-ALD) mode is used to improve the NH_3_ reactivity by plasma ignition in order to obtain AlN layers with suitable structural quality. PE-ALD grown AlN films on GaN typically exhibit a good epitaxial quality, giving rise to the formation of an interfacial 2DEG. However, the measured sheet electron density values are typically lower than in MBE-grown AlN with equivalent thickness and were found to depend on the deposition conditions [19] as well as on the deposited AlN thickness [21]. Moreover, significant oxygen incorporation is commonly observed in ALD grown AlN films, with the highest concentration at the film surface, probably due to exposure to the atmosphere after the deposition process [22,23,24]. The oxygen interdiffusion through the AlN layer occurs for few nanometers, but in the ultrathin layers the oxygen incorporation can reach also the interface region. In spite of the non-ideal quality of PE-ALD grown AlN, AlN/GaN transistors exploiting the interfacial 2DEG have been demonstrated using low-temperature (300 °C) PE-ALD on semi-insulating GaN [25]. Furthermore, AlN films deposited with such a low thermal budget proved to be effective passivation layers for AlGaN/GaN HETMs, leading to significant current collapse suppression and dynamic ON-resistance reduction without the use of a field plate [5,26].

In view of the above-discussed device applications, micro and nanoscale correlative studies of the structural, chemical and electrical properties of ALD grown ultra-thin AlN films on GaN would be highly desirable in order to assess their insulating properties. In particular, spatially resolved information on the lateral uniformity of current transport across the AlN thin films are currently missing and can be crucial to evaluate their suitability as tunneling barriers for vertical diodes or transistors.

In our work, the structural/compositional and electrical properties of 5 nm AlN films deposited by PE-ALD on GaN-on-sapphire substrates were investigated in detail by high resolution characterization techniques, i.e., transmission electron microscopy (TEM) combined with energy dispersive spectroscopy (EDS) and by conductive-atomic force microscopy (C-AFM). Chemically, the AlN layer is characterized by oxygen contamination, whose amount decreases moving from film surface to film/substrate interface. However, despite this contamination, the high degree of epitaxy at the AlN/GaN interface ensures the formation of a two-dimensional electron gas (2DEG), with a sheet charge density n_s_ ≈ 1.45 × 10^12^ cm^−2^. A uniform vertical current transport by electron injection through the AlN barrier was demonstrated by C-AFM current mapping. Furthermore, local current–voltage (I–V) measurements showed that the current transport follows the Fowler–Nordheim (FN) tunneling mechanism, with an average AlN/GaN barrier height of <Φ_B_> = 2.08 eV.

## 2. Materials and Methods

MOCVD n-type (~10^17^ cm^−3^) GaN grown on sapphire was used as substrate for the AlN deposition, which was carried out in a PE-ALD LL SENTECH reactor (Sentech, Instruments GmbH, Berlin, Germany) using trimethylaluminum (TMA) as the Al precursor (Air liquide, Catania, Italy) and NH_3_-plasma as co-reactant. A capacitively coupled plasma (CCP) source working through a 13.56 MHz RF-generator with a power of 200 W was used to generate the NH_3_-plasma reaction gas. Each ALD cycle consisted of 30 ms and 15 ms pulse times of TMA and NH_3_-plasma, respectively, alternated with a purging pulse of N_2_ to remove unreacted precursors and clean the deposition chamber. The deposition processes were performed at a temperature of 300 °C and a pressure of 20 Pa. The cycle number (60 deposition cycles) during the ALD process was established in order to achieve the desired thickness of 5 nm. Preliminary morphological analyses of the as-deposited AlN layers were carried out by tapping mode Atomic force microscopy (AFM), using a D3100 microscope (Bruker, San Francisco, CA, United States) with a Nanoscope V controller. Capacitance–voltage (C–V_g_) measurements on AlN film on GaN were performed using a mercury (Hg)-probe system. This method is highly beneficial because it provides a straightforward evaluation of the electrical behavior without any step of processing for the fabrication of capacitors. The capacitance was evaluated by measurements acquired between two front contacts, the smallest one consisting of a liquid Hg droplet with a controlled volume, and the other one consisting of a metal ring with a much larger area.

Cross-sectioned samples were prepared by focused ion beam (FIB) (SCIOS2 SEM+FIB dual beam manufactured by ThermoFisher, Brno, Czech Republic) and high resolution structural/chemical characterization by transmission electron microscopy (TEM), scanning transmission electron microscopy (STEM) and energy dispersion spectroscopy (EDS) performed with an aberration-corrected Titan Themis 200 microscope (ThermoFisher, Eindhoven, Netherlands).

Finally, the current transport across the 5 nm AlN layer on GaN was investigated at the nanoscale by performing conductive-atomic force microscopy (C-AFM) current mapping and local current–voltage (I–V_tip_) characterizations with diamond-coated Si tips, using DI 3100 AFM equipment (Bruker, San Francisco, CA, USA) with Nanoscope V electronics. Furthermore, local dC/dV vs. V_tip_ characteristics were collected by the scanning capacitance microscopy (SCM) module, using same diamond-coated Si tips, in order to evaluate the uniformity of the flatband voltage (V_FB_) for the tip/AlN/GaN heterostructure.

## 3. Results

Two representative AFM morphological images of the virgin GaN-on-sapphire substrate (a) and after the PE-ALD of AlN (b) are shown in Figure 1. The atomic terraces of the GaN surface remained clearly visible after deposition of the ultrathin (5 nm) AlN film, confirming the uniform and conformal coverage by the ALD process. The increased root mean square (RMS) roughness, from 0.28 nm of the bare GaN to 0.54 nm of the AlN coated surface, is due to the morphology of the deposited film.

A detailed structural and chemical investigation of the AlN film and its interface with GaN was carried out by high angle annular dark field (HAADF) STEM measurements combined with EDS, and by high-resolution TEM. Figure 2a reports a cross-sectional HAADF image, showing a very sharp atomic number Z-contrast between the AlN thin film and the GaN substrate. The Pt layer on top of AlN was used as protection during FIB sample preparation. The Z contrast allowed us to clearly visualize the interface between the AlN film and GaN and to precisely evaluate the film thickness, which coincided with the expected thickness of 5 nm. Furthermore, Figure 2b–f report the corresponding EDS chemical maps of Ga, N, Al, O and Pt, shown with different elemental combinations, from which the distribution of the different species in the analyzed stack could be clearly deduced. In particular, the presence of unintentional oxygen contamination within the deposited film was observed, similarly to what was reported in other papers on PE-ALD grown AlN layers [22,27]. In order to provide quantitative compositional information, the scan lines of the percent atomic concentrations for the Ga, Al, N and O elements in the stack are shown in Figure 2g. The interface between the GaN substrate and the deposited Al (O) N film, taken as the crossing point between the Ga and Al profiles, was indicated by the vertical dashed line at z = 0. A gradient in the oxygen concentration was clearly observed, with a decrease from ~70% at the film surface to ~15% at the interface with GaN. It should be noted that the evaluated oxygen percentage can be affected by artifacts, e.g., the natural oxidation of the cross-sectioned surface of the TEM lamella, which can justify the 10% oxygen content measured in the GaN region. The observed oxygen concentration gradient in the deposited Al (O) N film provides some indication of the possible sources of the oxygen contamination. In fact, oxygen incorporation during the PEALD growth, due to the presence of oxygen-based species activated by the plasma, would result in a uniform concentration within the deposited films. On the other hand, the decreasing oxygen concentration from the film surface to the interface with GaN is more consistent with oxygen incorporation occurring after the AlN layer growth by diffusion from the surface. This may happen either in the cooling step of the ALD process, in the presence or oxygen gas residuals in the chamber or, most probably, by exposure to the air atmosphere. In this respect, it can be important to evaluate the time-scale in which the oxidation occurs. The TEM-based chemical characterizations reported above are typically performed within one week from the sample deposition. Furthermore, analyses performed after longer times (approximately one month) exhibit the same qualitative behavior, suggesting that the samples do not undergo long time aging effects. As discussed later on in this paper, non-destructive electrical measurements (capacitance–voltage and C-AFM current maps) were also performed on the as-deposited samples, as well as after some days from the deposition. No significant variations of the electrical parameters (2DEG carrier density, homogeneity of injected current) were detected, suggesting that the oxidation occurred in the early stages, i.e., within one hour, from the exposure to the air.

In order to evaluate the lattice structure in the interface region between the deposited AlN film and the GaN substrate, a high-resolution TEM image is reported in Figure 3, showing an atomically abrupt AlN/GaN interface, with a perfectly epitaxial alignment in the first AlN atomic layers. The AlN film exhibited the hexagonal stacking of the wurtzite structure characteristic of GaN material in the interfacial region, whereas a deviation from this stacking order could be observed at a distance of ~1 nm from the interface. The measured values of the (0002) plane distances were 0.2573 nm for GaN and 0.2472 nm for AlN. The X-ray reference standards for the two crystals were 0.259 nm for GaN (0002) plane distances (JCPDS card 02-1078) and 0.249 nm for the AlN (0002) plane distances (JCPDS card 25-1133). Hence, the determined values are in good agreement with the references.

Noteworthily, in spite of the very large oxygen content in the surface region of this 5 nm thick film, the epitaxial arrangement of AlN with GaN was preserved in the near interface region, guaranteeing the generation of a 2DEG, as confirmed by Hg-probe capacitance–voltage (C–V_g_) measurements. In particular, Figure 4a shows a typical C–V_g_ curve measured at 100 kHz frequency. The 2DEG sheet carrier density n_s_ as a function of V_g_ (see Figure 4b) was calculated by integration of the C–V_g_ and subtracting the contribution associated with the GaN substrate doping. The relatively low values of n_s_ (0) = 1.45 × 10^12^ cm^−2^ and of the 2DEG pinch-off voltage (V_po_ = −1.8 V) can be explained by the small thickness (~1 nm) of epitaxial AlN as compared to the total of deposited thickness (5 nm), as deduced from TEM analyses.

Then, the vertical current transport across the ultra-thin AlN barrier layer on GaN was investigated at the nanoscale by C-AFM. Figure 5a,b show the typical morphological image and the corresponding current map collected by scanning the diamond tip on the sample surface, while applying a bias V_tip_ = 5 V with respect to a large area electrode deposited on top of AlN, as schematically illustrated in the inset of Figure 5c. Quite uniform electron injection from the interfacial 2DEG through the thin AlN layer could be deduced from this map. To gain further insight in the current transport mechanism through the AlN film, local current–voltage (I–V_tip_) measurements were performed by displacing the tip on an array of 5 × 5 positions. The collected I–V_tip_ curves exhibited a rectifying behavior, with a very low current level under reverse (negative) polarization of the tip, and an exponential increase of the current for positive bias values V_tip_ > 3 V.

As a complementary analysis to C-AFM current mapping, Figure 6 reports a set of dC/dV vs. V_tip_ characteristics measured on an array of 5 × 5 positions of the diamond tip on the AlN surface, using the SCM setup (as schematically depicted in the inset of the same figure) [28]. The nanoscale differential capacitance for the AlN/GaN heterostructure depends on the local AlN barrier thickness and dielectric constant, as well as on the GaN doping and interfacial 2DEG density. Furthermore, the peak voltage in the curves corresponds to the local flatband voltage (V_FB_) of the diamond tip/AlN/GaN metal/insulator/semiconductor system, which is related both to the tip/semiconductor workfunction difference and to charges in the AlN Film. The fact that the local dC/dV curves measured at different surface positions all overlapped each other demonstrates a high lateral uniformity of the dielectric properties of the AlN film, as well as of the carrier density at the interface with GaN. In particular, the determination of the local flatband voltage value (V_FB_ = 0.3 V) is important to gain deeper insight in the current transport mechanism through the AlN layer, starting from the local I–V_tip_ characteristics in Figure 5c.

The exponential dependence of the current in these characteristics was modelled by the Fowler–Nordheim (FN) tunneling mechanism, typically employed to describe electronic transport across thin insulating films [29]. According to the FN equation, the current tunneling through the triangular barrier can be expressed as:(1)$$I=\frac{q^{3}mE^{2}}{8\pi h\Phi_{B}m_{ox}}exp\left[-\frac{8\pi \sqrt{2m_{ox}\Phi_{B}^{3}}}{3qh}\frac{1}{E}\right]$$
where E = (V_tip_−V_FB_)/d is the local electric field applied by the tip (d = 5 nm being the barrier layer thickness), Φ _B_ is the energy barrier at the AlN/GaN interface, q is the electron charge, h is the Boltzmann constant and m_ox_ is the effective tunneling mass of the thin barrier layer. Since the deposited Al (O) N layer featured a large oxygen concentration gradient (from ~70% at the surface to ~15% at the interface with GaN), the tunneling effective mass should be an intermediate value between the AlN and Al_2_O_3_ ones. However, quite similar effective mass values (m_ox_ ≈ 0.4 m_0_, m_0_ being the free electron mass) were reported in the literature for Al_2_O_3_ [30] and AlN [31]. Hence, the m_ox_ ≈ 0.4 m_0_ value was considered in Equation (1).

Figure 7a shows the FN plot (i.e., ln (I/E^2^) as a function of 1/E) for one of the forward I–V curves in Figure 5c. At high electric field values, the linear fit of this curve exhibited a correlation factor R very close to unity, indicating excellent agreement of the data with the FN model. A barrier height Φ _B_ = 2.01 eV was extracted from the slope of the fit. Furthermore, Figure 7b shows the histogram of Φ _B_ values obtained by performing the same fitting procedure on each of the I–V curves in Figure 5c. An average barrier height <Φ_B_> = 2.08 eV with a standard deviation of ±0.19 eV was evaluated from this distribution. Noteworthly, these values were in reasonable agreement with the calculated ones for the AlN/GaN conduction band offset ΔE_C_ (from 2.1 to 2.7 eV) [32,33]. This result indicates that in spite of oxygen incorporation in the PE- ALD grown thin film, the current injection behavior is ultimately determined by the high quality epitaxial interface between AlN and GaN.

## 4. Conclusions

In conclusion, ultra-thin (5 nm) AlN films with uniform and conformal morphology to the GaN substrate were grown by PE-ALD at a temperature of 300 °C. Structural investigation by high-resolution TEM demonstrated an atomically abrupt interface between GaN and AlN, with a perfect epitaxial alignment of the first atomic layers. However, a deviation from the wurtzite stacking was observed at a distance of ~1 nm from the interface. Chemical analysis by EDS profile demonstrated oxygen contamination of the AlN layer. This contamination, attributable to the atmosphere exposure after the ALD process, is mainly observed on the surface region, whereas it reduces to a rather low value (~10–15%) at the interface. The epitaxial interface results in the formation of a 2DEG with a sheet carrier density of 1.45 × 10^12^ cm^−2^. The current map by C-AFM demonstrated a uniform vertical current transport by the electron injection from the interfacial 2DEG through the AlN barrier layer. The current transport was identified with the Fowler–Nordheim (FN) tunneling mechanism, and an average barrier height value of <Φ_B_> = 2.08 eV was estimated, in good agreement with the expected AlN/GaN conduction band offset.

This work is useful for future application of AlN thin films by ALD as a tunneling barrier for GaN-based vertical devices. The incorporation of oxygen inside the AlN layer is certainly a crucial aspect for the possible effects on the structural and electrical properties of the AlN films. For this reason, it needs to be controlled to improve the AlN stability, for example by optimization of the ALD process, the cooling step and/or the post-deposition treatment. However, the ALD process, despite the low deposition temperature, guarantees an optimal epitaxy, mainly in the first atomic layers, which is appropriate for the 2DEG formation and highly uniform current injection.

## Figures and Tables

**Figure 1 nanomaterials-11-03316-f001:**
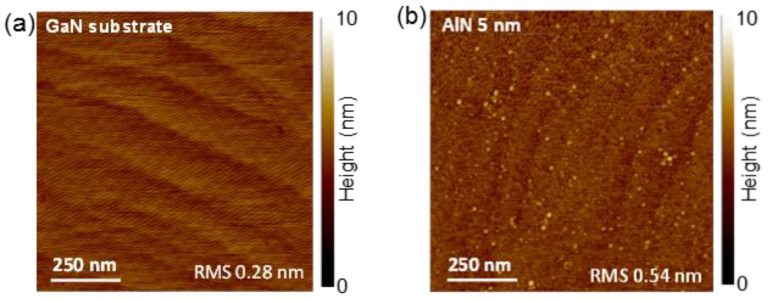
Morphological images of the virgin GaN substrate (**a**) and after PE-ALD of 5 nm thick AlN film (**b**).

**Figure 2 nanomaterials-11-03316-f002:**
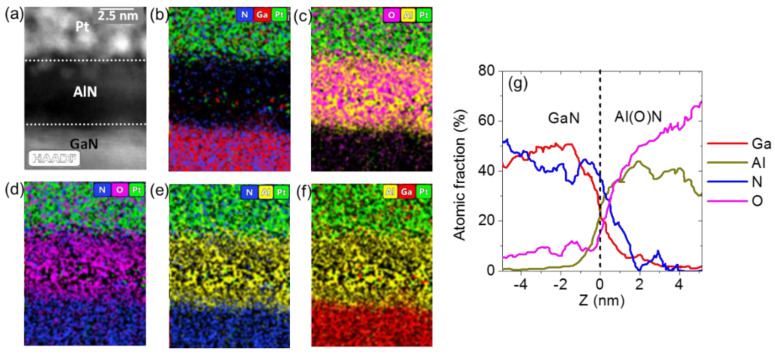
(**a**) Cross-sectional HAADF-STEM image of the 5 nm AlN film on GaN. (**b**–**f**) Corresponding EDS chemical maps of Ga, N, Al, O and Pt, shown with different elemental combinations, from which the distribution of the different species in the analyzed stack can be clearly deduced. (**g**) EDS scan lines of the percent atomic concentrations for the Ga, N, Al and O species.

**Figure 3 nanomaterials-11-03316-f003:**
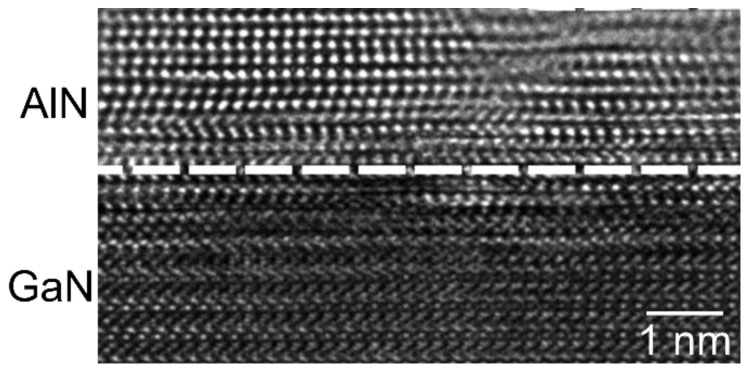
High resolution TEM image of the AlN/GaN interface region.

**Figure 4 nanomaterials-11-03316-f004:**
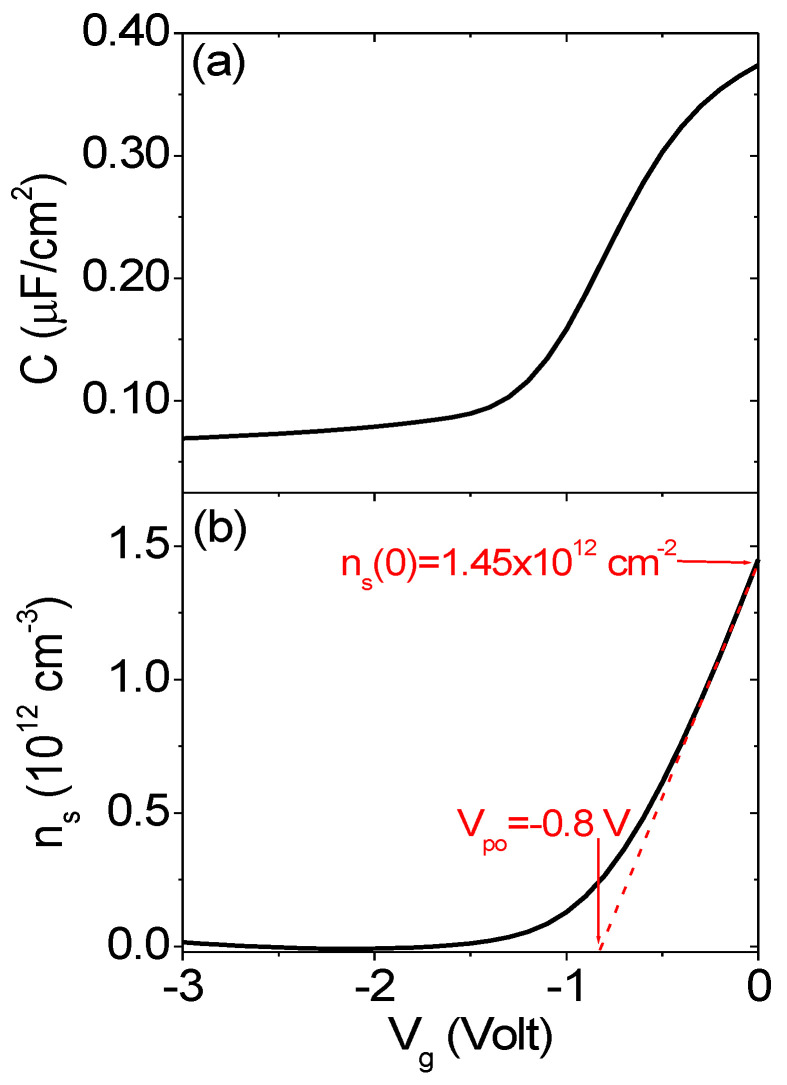
(**a**) Capacitance–voltage (C−V_g_) curves measured by the Hg−probe setup on the AlN/GaN heterostructure. (**b**) 2DEG sheet carrier density n_s_ (cm^−2^) as a function of V_g_ obtained by integration of the C–V_g_ curve and subtraction of the GaN doping contribution. The carrier density at V_g_ = 0 and the 2DEG pinch-off bias (V_po_) are indicated.

**Figure 5 nanomaterials-11-03316-f005:**
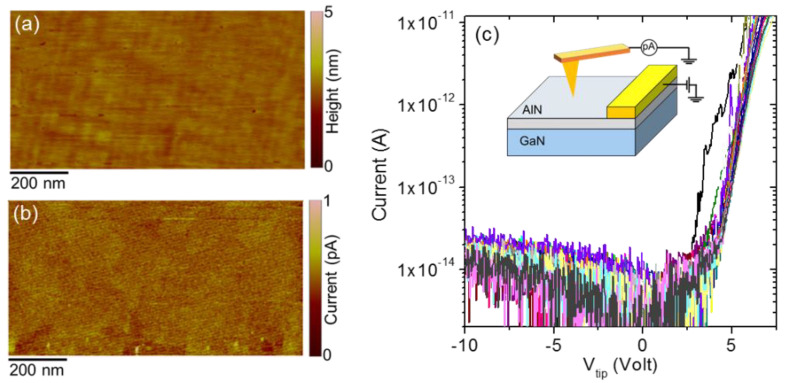
(**a**) Morphology and (**b**) vertical current map on 5 nm AlN film on GaN measured by C-AFM at a tip bias V_tip_ = 5 V. (**c**) Local I–V curves measured on an array of 5 × 5 positions of the diamond-tip on the AlN surface. The experimental configuration for C-AFM measurements is schematically illustrated in the insert of panel (**c**).

**Figure 6 nanomaterials-11-03316-f006:**
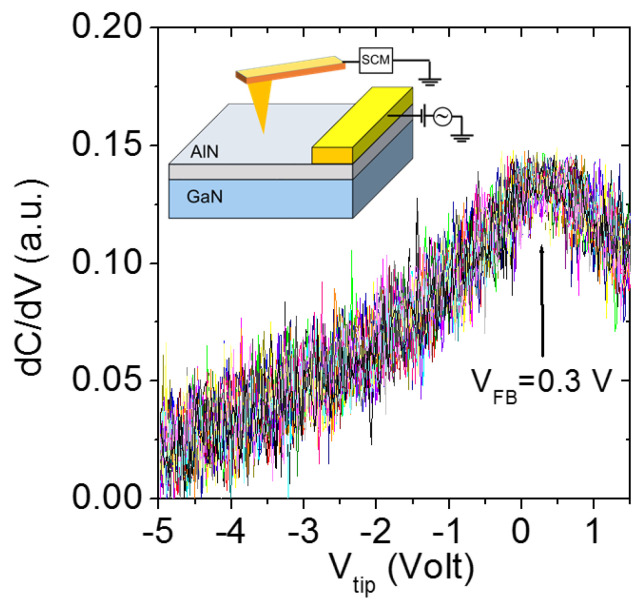
Local dC/dV–V_tip_ curves measured on an array of 5 × 5 positions of the diamond-tip on the AlN surface. The flatband voltage V_FB_ = 0.3V for the diamond tip/AlN/GaN heterostructure is indicated. The experimental configuration for SCM measurements is schematically illustrated in the inset.

**Figure 7 nanomaterials-11-03316-f007:**
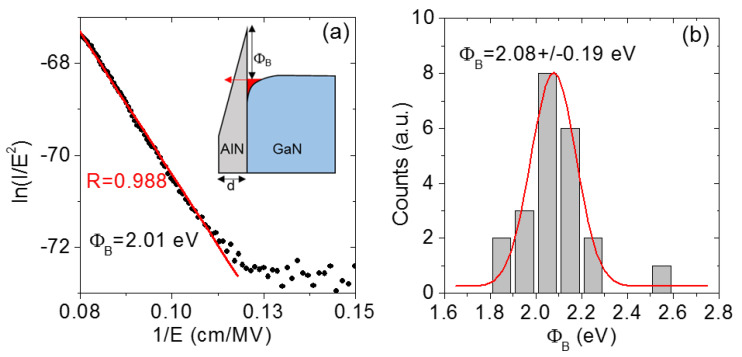
(**a**) Fowler–Nordheim plot of a forward bias I–V curve measured on AlN (5nm)/GaN and results of the linear fit. The schematic band-diagram of the heterostructure under forward polarization is reported in the insert. (**b**) Histogram of the barrier height values obtained by fitting of the I–V curves acquired at different surface positions.

## Data Availability

The data that support the findings of this study are available from the corresponding author upon reasonable request.

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
