# Peer review of "Highly Homogeneous Current Transport in Ultra-Thin Aluminum Nitride (AlN) Epitaxial Films on Gallium Nitride (GaN) Deposited by Plasma Enhanced Atomic Layer Deposition"

_nanomaterials, 2021, doi:10.3390/nano11123316_

Round 1
Reviewer 1 Report
In this paper the authors report on the properties of AlN epitaxial thin films grown on GaN by plasma enhanced ALD. The first half of the manuscript is very convincing and clearly written. However, the second half of the paper, especially the discussion on the current transport mechanism isn't clear. Definitely, this relys to a great extend on the fact that figures and analysis are missing. Page 8, lines 265-292: The mentioned Figures 6a and 6b as well as Fig. 4c don't exist in the manuscript uploaded for review. I'm wondering from which document this part was copied and pasted. In the letter accompanying the revised version, the authors might clearly demonstrate that the revised version isn't a simple copy of an already published paragraph.
Author Response
Reviewer 1
In this paper the authors report on the properties of AlN epitaxial thin films grown on GaN by plasma enhanced ALD. The first half of the manuscript is very convincing and clearly written. However, the second half of the paper, especially the discussion on the current transport mechanism isn't clear. Definitely, this relies to a great extent on the fact that figures and analysis are missing. Page 8, lines 265-292: The mentioned Figures 6a and 6b as well as Fig. 4c don't exist in the manuscript uploaded for review. I'm wondering from which document this part was copied and pasted. In the letter accompanying the revised version, the authors might clearly demonstrate that the revised version isn't a simple copy of an already published paragraph.
We thank the Reviewer for the positive comments on our manuscript. As he/she correctly observed, a figure (Fig.7(a) and (b)) was not included by mistake in the submitted version of the manuscript. We apologize for this inconvenience. Furthermore, Fig.5(c) must be considered instead of Fig.4(c).
We hope that, after solving these issues, the discussion of the current transport mechanism became clearer.
We obviously confirm that the present version of the manuscript is unpublished and is currently submitted for publication in Nanomaterials.
Reviewer 2
The article “Highly homogeneous current transport in ultra-thin aluminium nitride (AlN) epitaxial films on gallium nitride (GaN) deposited by plasma enhanced atomic layer deposition” is devoted to problem of creation good passivating dielectric on GaN. Although the authors actually obtained a film of aluminum oxynitride, they nevertheless solved the problem. The article contains new experimental data and may be published after minor revision.
1) Maybe the authors should mention right in the title AlN(O), not pure AlN.
2) There are no Fowler-Nordheim curves in Figure 6. Apparently, the authors wanted to show Figures 7a and 7b but forgot to place them in the article. This is the main point.
3) There is other negligence too. So on page 3 line 99 degrees should be Now it looks like 1017 cm-3.
Minor revision.
We thank the reviewer for his positive evaluation of our paper and the useful suggestions, aimed to further improve its quality. We addressed all the comments as follows:
- The deposited AlN thin films, is not actually a full AlN(O) layer, but it consists of a AlN epitaxial layer, containing oxygen contamination. This contamination is not equally distributed within the films, but occurring because of a diffusion process from the surface, it is more present at the top of the AlN layer, while the first 2 nm are completely epitaxial and possess a plane distance (0.2472 nm) in fully agreement with the theoretical value (0.249 nm) of AlN single crystal. We understand the referee’s comment on the possibility to change the title, but our feeling is that the replacement of AlN with AlN(O) could be misleading and not compatible with the epitaxial crystalline nature of the AlN/GaN interface. Moreover, since already in the abstract it is clearly indicated that oxygen contamination is present, we believe that our real finding is not hidden to potential readership.
- As correctly observed by the reviewer, Fig.7(a) and (b) were not included by mistake in the submitted version of the manuscript. We apologize for this inconvenience. The figures have been now included in the revised version of the paper.
3. At page 3 of the revised paper, the correct notation 1017 cm-3 was now used

Reviewer 2 Report
Referee Report
The article “Highly homogeneous current transport in ultra-thin aluminium nitride (AlN) epitaxial films on gallium nitride (GaN) deposited by plasma enhanced atomic layer deposition” is devoted to problem of creation good passivating dielectric on GaN. Although the authors actually obtained a film of aluminum oxynitride, they nevertheless solved the problem. The article contains new experimental data and may be published after minor revision.
- 1) Maybe the authors should mention right in the title AlN(O), not pure AlN.
- 2) There are no Fowler-Nordheim curves in Figure 6. Apparently, the authors wanted to show Figures 7a and 7b but forgot to place them in the article. This is the main point.
- 3) There is other negligence too. So on page 3 line 99 degrees should be Now it looks like 1017 cm-3.
Minor revision.
Author Response
Reviewer 2
The article “Highly homogeneous current transport in ultra-thin aluminium nitride (AlN) epitaxial films on gallium nitride (GaN) deposited by plasma enhanced atomic layer deposition” is devoted to problem of creation good passivating dielectric on GaN. Although the authors actually obtained a film of aluminum oxynitride, they nevertheless solved the problem. The article contains new experimental data and may be published after minor revision.
1) Maybe the authors should mention right in the title AlN(O), not pure AlN.
2) There are no Fowler-Nordheim curves in Figure 6. Apparently, the authors wanted to show Figures 7a and 7b but forgot to place them in the article. This is the main point.
3) There is other negligence too. So on page 3 line 99 degrees should be Now it looks like 1017 cm-3.
Minor revision.
We thank the reviewer for his positive evaluation of our paper and the useful suggestions, aimed to further improve its quality. We addressed all the comments as follows:
- The deposited AlN thin films, is not actually a full AlN(O) layer, but it consists of a AlN epitaxial layer, containing oxygen contamination. This contamination is not equally distributed within the films, but occurring because of a diffusion process from the surface, it is more present at the top of the AlN layer, while the first 2 nm are completely epitaxial and possess a plane distance (0.2472 nm) in fully agreement with the theoretical value (0.249 nm) of AlN single crystal. We understand the referee’s comment on the possibility to change the title, but our feeling is that the replacement of AlN with AlN(O) could be misleading and not compatible with the epitaxial crystalline nature of the AlN/GaN interface. Moreover, since already in the abstract it is clearly indicated that oxygen contamination is present, we believe that our real finding is not hidden to potential readership.
- As correctly observed by the reviewer, Fig.7(a) and (b) were not included by mistake in the submitted version of the manuscript. We apologize for this inconvenience. The figures have been now included in the revised version of the paper.
3. At page 3 of the revised paper, the correct notation 1017 cm-3 was now used
Round 2
Reviewer 1 Report
Thank you for the revision. The revised version might be published in the present form.